# The Role of Gamma Delta T Cells in Autoimmune Rheumatic Diseases

**DOI:** 10.3390/cells9020462

**Published:** 2020-02-18

**Authors:** Ilan Bank

**Affiliations:** Rheumatology Unit, Autoimmunity Center, Sheba Medical Center, Tel-Hashomer 52621, Israel; ibank@tauex.tau.ac.il

**Keywords:** gammadelta T cells, rheumatoid arthritis, systemic lupus erythematosus, systemic sclerosis, ankylosing spondylitis, juvenile idiopathic arthritis

## Abstract

Autoimmune rheumatic diseases (ARDs), affecting ~1–1.5% of all humans, are associated with considerable life long morbidity and early mortality. Early studies in the 1990s showed numerical changes of the recently discovered γδ T cells in the peripheral blood and in affected tissues of patients with a variety of ARDs, kindling interest in their role in the immuno-pathogenesis of these chronic inflammatory conditions. Indeed, later studies applied rapid developments in the understanding of γδ T cell biology, including antigens recognized by γδ T cells, their developmental programs, states of activation, and cytokine production profiles, to analyze their contribution to the pathological immune response in these disorders. Here we review the published studies addressing the role of γδ T in the major autoimmune rheumatic diseases, including rheumatoid arthritis, juvenile idiopathic arthritis, ankylosing spondylitis, systemic lupus erythematosus and scleroderma, and animal models thereof. Due to their unique properties spanning adaptive and innate immune functions, the ever deeper understanding of this unique T cell population is shedding new light on the pathogenesis of, while potentially enabling new therapeutic approaches to, these diseases.

## 1. Introduction

In the mid 1980s, the previously elusive nature of the T cell receptor (TCR) expressed by CD4^+^ and CD8^+^ major histocompatibility complex (MHC) restricted T cells had just been established to be encoded by rearranging α and β TCR gene [1,2]. However, the serendipitous discovery of a third rearranging gene, termed γ, in a murine clone of cytotoxic αβ T cells, confounded by absent expression of a protein encoded by this gene, raised questions relating to the role of this newcomer [3]. These were resolved in 1986, when two papers revealed human thymocyte derived CD4^−^CD8^−^ T cell clones and peripheral blood T cell clones expressing a second TCR composed of two polypeptide chains associated with the CD3 molecule, one of which was encoded by the “mysterious” γ gene, and the second later shown to be encoded by a fourth TCR gene, δ [4,5,6].

Since those early years, γδ T cells have been shown to be prototypes of “unconventional” T cells. Their unconventionality, is exemplified by the opportunity they present for broadening the “universe” of antigens that can be recognized by T cells. Thus, as opposed to MHC restricted CD4^+^ and CD8^+^ αβ T cells, γδ TCR do not recognize peptide antigens within classical MHC molecules. Rather, their T cell receptors mediate direct recognition of non peptidic molecules currently known to include cell surface expressed butyrophilins (whose recognition and stimulatory properties are enhanced by intracellular low molecular weight phospho-antigens), phycoerythrine, glycosides, t-RNA synthetase, and other intracellular enzymes, heat shock proteins, the non-classical MHC like molecules CD1, MR1, and endothelial protein C receptor (EPCR) [7]. Furthermore, different subsets of γδ T cells distinguished by the V genes used in their TCR may have evolved to recognize different antigens and presenting molecules. For example, human γδ TCR that use the Vγ9 and Vδ2 genes (Vγ9Vδ2^+^ γδ T cells) recognize butyrophilin 3A1 to which a phosphoantigen has bound intracellularly on target cells [8]. By contrast, other butyrophilins, as well as CD1d, may serve as antigenic targets of the second major subset of human γδ T cells, namely those using the Vδ1 gene [9,10]. This unusual antigenic repertoire however, is only now beginning to be unraveled, and is likely to be greatly expanded and detailed in the future. In addition, γδ T cells follow unique intra- and extra-thymic pathways of functional maturation [11,12]. Thus, for example, imprinting of subsets of γδ T cells resulting in an innate ability to produce the potent proinflammatory interleukin (IL)-17, in the absence of TCR activation, takes place during thymic maturation [13]. This contrasts with the requirement for antigenic encounters in peripheral lymph nodes, in order for conventional naïve αβ T cells to acquire effector functions. An additional distinguishing feature is the affinity of certain subsets of γδ T cells, to establish residence, directly after exit from the thymus, in specific peripheral tissues, most prominently along mucosal and dermal epithelial surfaces [12]. Furthermore, the specific tissue is dictated by V genes of the γδ TCR [12]. These features, together with other broad functional properties overlapping those of innate and adaptive immune cells i.e., secretion of a wide spectrum of cytokines potential including tumor necrosis factor (TNF)α type I and II cytokines, to mediate potent cytotoxicity, help for B cells, immune regulatory potential, and even regulation of tissue metabolism, positions these cells to serve a unique and non redundant role within the immune system and has been the subject of recent excellent reviews, highly recommended to the reader [11,12,14]. Furthermore, γδ T cells were shown to undergone major perturbations in the context of infectious, autoimmune, and malignant diseases [15].

In the context of disease conditions, interest of the scientific community has primarily focused on γδ T cells in cancer, encompassing their contradictory ability to either control or enhance malignancy [16]. Only ~100 published studies have studied γδ T cells in the context of autoimmune rheumatic disorders, which represent a significant health burden affecting 1–1.5% of the global human population with considerable morbidity, and in some diseases increased [17]. In this comprehensive review, we summarize research from publications found in PubMed in which the major autoimmune rheumatic disorders (ARDs) (rheumatoid arthritis (RA), juvenile idiopathic arthritis (JIA), systemic lupus erythematosus (SLE), ankylosing spondylitis (AS) and systemic sclerosis (SSc)) were addressed with reference to γδ T cells. Experimental animal models representing these diseases, in which the contribution of γδ T cells was studied, are also described. When considered in total, it appears that γδ T cells play an important and unique role in these diseases. Furthermore, given the increasing understanding of the underlying principles governing γδ T cell recognition and physiology, these and future studies are contributing to a wider comprehension of the pathogenesis of ARDs, and may point to new ways of treating these devastating conditions.

## 2. Rheumatoid Arthritis

### 2.1. Numerical Alterations of γδ T Cells in RA

In an early study, a decrease of CD4^-^CD8^-^ (mostly γδ T cells) in peripheral blood (PB) of RA patients relative to healthy controls (HC) was noted (1.38 ± 1.08% vs. 3.23 ± 2.12%, *p* < 0.05], whereas these cells were increased in synovial fluid (SF) of patients [18]. Similarly, a decrease in PB in both RA and psoriatic arthritis (PsA) patients relative to HC was found in a different cohort [19]. However, in another study, although RA in young (40.9 ± 7.5 years) was associated with higher levels of PB γδ T cells than in old (76.1 ± 4.9 years) patients, their percentage was not different from age matched controls [20]. Likewise, while increased γδ T cells were noted in the lamina propria in the intestinal mucosa (mean 5.5%, range 2–12%) in rheumatoid factor (RF) positive patients (*n* = 8) compared with RF negative RA patients and a disease control group (*n* = 15, mean 2%, range 0.5–5%; *p* < 0.01) similar changes were not detectable in PB [21]. In yet another study, the percentages (mean ± SEM = 6.3 ± 0.8%, *n* = 22) and absolute numbers (70 ± 11/microliters, *n* = 22) of γδ T cells in PB from RA patients were not different from those of 22 age-matched HC (7.5 ± 0.9%, 81 ± 17/microliters, respectively) [22]. Interestingly however, among a cohort of 24 RA patients, γδ T-cell levels were likewise not significantly different between controls, 4.46 ± 1.36%, gold salt treated (GST, 6.88 ± 1.73%), and total RA patients (2.73 ± 0.55%), but 42% of the GST treated group had γδ T-cell levels higher than the entire untreated RA group [20]. Finally, as opposed to these studies predominantly showing either unaltered or decreased levels of γδ T cells in the PB of RA patients, a single study reported 10 patients with RA in whom γδ T cells were 5.5% ± 4.38 (mean ± s.d.), which was significantly increased as compared with 22 healthy subjects (2.09 ± 1.01, *p* < 0.001) [23]. 

With respect to subsets of γδ T cells, one study reported that in early RA (> 6 months (m) < 8 m disease duration) the percentage of Vγ9Vδ2^+^ T cells in the PB was the same as controls. Their percentage in synovium, however was higher than in PB of patients and controls. These cells also expressed high levels of human leukocyte antigen (HLA)-DR and CD86 [24]. Concurring with this, the total percentage of Vγ9Vδ2 T cells was the same as controls among another group of early RA patients, most of whom were anti citrulline peptide antibody (ACPA) positive. However, among these, there was an increase of Vγ9Vδ2 T cells bearing a terminal effector memory CD27^-^CD45RA^+^ phenotype (TEMRA) and a decrease of naïve CD27^+^CD45RA^+^ cells [25]. Contrasting with these results, among 19 adults with early active RA, 80% of whom were RF^+^ or anti-cyclic citrullinated peptide (CCP) ^+^ and on no current steroid treatment, Vγ9Vδ2 T cells and regulatory T cells (Tregs) were lower, whereas the total percent of γδ T cells was same as in HC [26]. Likewise, among 68 patients with RA (not necessarily designated as early RA), 21 with osteoarthritis (OA) and 21 HC, the percent of γδ T cells in PB was found to be significantly lower in the RA patients, and the percent of Vδ2^+^ T cells in PB was also decreased in RA relative to OA and HC. By contrast, in SF and synovial tissue Vδ2^+^ T cells were increased (~5.9% vs. 1.2%). Interestingly, anti tumor necrosis factor (TNF)α treatment was associated with increased levels of Vδ2^+^ cells in the periphery [27]. Similarly, Lamour found that the total γδ T cell percentage decreased relative to HC, and that the Vδ2^+^ subset was decreased relative to the Vδ1^+^ subset. Furthermore, human leukocyte antigen (HLA)-DR increased during active disease on γδ T cells of RA patients [28]. Thus, in RA, the PB γδ T cell subset expressing the common Vγ9 and Vδ2 combination in the TCR (Vγ9Vδ2 T cells), appears to be unchanged or decreased—in particular in advanced phases of the disease—and bears stigmata of having been activated during the disease process. Furthermore, since PB may be relatively depleted of Vγ9Vδ2 T cells, whereas synovial Vγ9Vδ2 T cells are relatively expanded in synovium relative to PB, transmigration of this subset to the site of inflammation appears to be one mechanism of accumulation of γδ T cell in the rheumatoid synovium. 

In contrast to the findings in most of these studies, showing normal or decreased percentages of γδ T cells in the PB of RA patients, large expansions of these cells can be found in patients with RA and large granular lymphocyte (LGL) proliferation of γδ T cells. Thus, in one study, 3.6% of patients with RA had >10% LGL (CD3^+^CD56^+^) in the PB. These patients were not clinically distinct, other than developing cytopenias, but significantly more were under anti TNFα therapy and ~60% of the LGL expansions were γδ T cell clones [29]. Furthermore, among 14 patients with γδ T cell-LGL leukemia, (11 men and three women), six had a history of RA. Eight of 12 patients had a CD4^–^CD8^–^ phenotype, and 4 had a CD4^–^CD8^+^ phenotype. In this study, patients with γδ T-LGL leukemia were more likely to have RA than those with other forms of LGL leukemia (*p* = 0.04) and median overall survival for the six patients with RA was 209 months, compared to 62 months for eight patients without RA (*p* = 0.7) [30]. In another study, however, while 20% of patients with γδ LGL had RA (4/20), a similar proportion of 169 patients with αβ LGL also developed RA [31].

### 2.2. γδ T Cells in Rheumatoid Synovium

Whereas synovia from patients with OA, SLE and joint trauma did not show an increased presence of γδ T cells, these cells were increased in a subset of RA patients. RA patients with γδ T cell infiltrates in the synovium had an increased tissue inflammation score compared to RA synovia with few γδ T cells [18.6 ± 5.8 versus 11.6 ± 4.2, *p* < 0.05] [32]. In another study, among 23 rheumatoid synovial membranes, using immunohistology and monoclonal antibodies (mAb), the majority showed only limited staining for δ-chain antibodies, with 20 of the 23 tissues appearing to have less than 1% of T lymphocytes expressing δ chains. Nevertheless, three tissues stained extensively for both δ (all γδ T cells) and δ TCS1 (Vδ1^+^) in particular areas of the section. In these areas, small perivascular lymphocytic aggregates appeared to be composed mainly of γδ T cells [33]. In addition, the expression of CD16 was reduced, and HLA-DR increased in synovial γδ T cells in RA patients [34]. These findings indicate that γδ T cells participate in the inflammatory process occurring in the synovium in RA and express an activated phenotype.

### 2.3. TCR Gene Expression

A study of synovial membrane lymphocytes from the RA patients, which confirmed a selective expansion of γδ T cells (8.8% in synovial membrane versus 4% in PB) also found, by immunohistochemical studies, that the TCR of the γδ T cells was unusual inasmuch as most γδ T cells did not express Vδ or Vδ genes, that predominate in PB [35]. Further analysis of synovial T cells, using a mAb (B18) specific for Vγ8 revealed, indeed, that in PB of healthy persons, only 6 ± 5% and only 1 of 35 γδ T cell clones were Vγ8^+^, whereas the B18^+^ subset was a dominant γδ T cell population among intraepithelial lymphocytes (IEL) derived from the human intestine (74 ± 29%, *p* < 0.002), and in the SF of patients with RA (21 ± 18%, *p* < 0.05 compared with normal PB). Furthermore, the B18^+^ subset was more frequent among IL-2-expanded γδ T cells (42 ± 20%) derived from synovial tissue than among IL-2-expanded cells derived from HC PB (*p* < 0.002) and PB from RA patients (*p* < 0.02). All B18^+^ clones (*n* = 7) expressed mRNA for Vγ8 together with mRNA for Vδ1 (*n* = 5) or mRNA for Vδ3 (*n* = 2). Thus, γδ T cells expressing Vγ8, together with mainly Vδ1, form a major γδ T cell subset among the IEL of the gut and a highly frequent subset in the synovial tissue of patients with RA [36]. In another study, reverse transcriptase-polymerase chain reaction (RT-PCR), in conjunction with nucleotide sequencing, revealed a frequent usage of the Vγ3 gene segment in RA synovial fluid mononuclear cells (SFMC) which was rare in PBMC of healthy individuals, where the Vγ9 gene predominated. The Vγ3 gene in RA SFMC showed no conserved junctional sequence and Vγ3 expressing clones were non-reactive to mycobacterium tuberculosis, as opposed to the Vγ9^+^ clones [37]. Others, using PCR to amplify TcR γ- and δ-chain transcripts, found SFMC expressing TCR γ-chain transcripts which used the same set of Vγ genes as peripheral blood mononuclear cells (PBMC). The majority of patients expressed a restricted SMC Vδ-chain repertoire biased towards Vδ1. Vδ2 mRNA transcripts were also detected, albeit at low levels, in some patients [38]. Interestingly, the level of expression of the 4 Vγ gene family members was determined by PCR, and 509 cDNA clones were derived from 8 SF and one PB sample from 2 patients with RA and one patient with JIA, subcloned and sequenced. Disproportionate expression of a subpopulation of TCR γ mRNA transcripts were found in each patient and some of these transcripts were expressed by T cells found in both joints, consistent with a common antigen driven response in the joints [39]. In addition, in contrast to control PBMC, Vδ1 chain cDNA derived from PBMC of three patients showed a strong bias towards usage of the same V-joining (J) combination and junctional region sequences, although the specific sequences were unique in each patient, whereas oligoclonality of the Vδ1 chain was less marked in SFMC of two of these patients and absent in SFMC of the other patients. For Vδ2, oligoclonality was detected in PBMC of two patients. In SFMC of a single patient, a dominant Vδ2 transcript was detected that utilized the Jδ2 segment, which was rarely expressed in the normal TCR repertoire. These results indicate in vivo clonal expansion of Vδ1- and Vδ2-expressing γδ T cells in the PB of RA patients contrasting with a synovial T cell infiltrate which consists largely of polyclonally expanded γδ T cells, but showing clonal dominance in some patients [40]. Moreover, it appears that γδ T cells may expand in RA synovium to consist a unique population characteristically enriched in Vδ1^+^ T cells and often co-expressing Vγ8 and Vγ3 genes, which suggests they recognize MR1 [41]. In summary, the synovium in RA contains γδ T cells with a polyclonal repertoire, although sometimes containing oligoclonal expansions common to different joints. Synovial Vγ9Vδ2^+^ T cells, which consist of the predominant phenotype in healthy PB, may be expanded in synovium relative to the patient’s PB, but usually form a less prominent component of the synovial γδ T cell infiltrate. The infiltrate may include less common types of γδ T cells, but is usually enriched for Vδ1^+^ T cells that use unusual Vγ genes, some of which have been associated with reactivity with non classical MHC like molecules. Together, these findings suggest that synovial γδ T cells may be selected by specific antigens found in the synovium. 

### 2.4. Functions of γδ T Cells in RA

Vγ9Vδ2 T cells isolated from patients with early RA were found to be capable of presenting peptide antigens to CD4^+^ T cells. In support of this, they expressed high levels of HLA-DR and CD86, molecules involved in antigen presentation, and characteristic of antigen presenting cells [24]. In addition, IFNγ (~50%), TNFα (~40%), and IL-17 (~3.7%) were demonstrated to be produced by the indicated percentages of RA synovial Vγ9^+^ T cells. Furthermore, RA SF Vδ2^+^ T cells expressed high levels of C-X-C motif chemokine receptor (CXCR) 3 and C-C motif chemokine receptor (CCR) 5 that were upregulated by TNFα in a nuclear factor (NF)-κb dependent pathway, and migrated to RA SF more efficiently than HC and OA derived cells. Vδ γδ T cells, however, had lower levels of the chemokine receptors [27]. In another study, among 22 γδ T cell clones obtained from the SF and PB of one patient with inflammatory arthritis (and compared to 26 αβ TCR^+^ T cell clones of the same and different patients), IFNγ was produced by 82% and IL-4 by 77% of the 22 γδ T cell clones whereas IL-10 was not. The mean levels of IL-4 were lower for clones derived from SF. Thus, the most common pattern was a γδ Th1-like pattern, primarily found in SF derived Vδ1^+^ clones. A γδ Th0-like pattern (balanced production of both IFNγ and IL-4), a γδ Th1 pattern (IFNγ alone) and a γδ Th2 pattern (IL-4 alone) were also found. These three patterns were also seen in PB Vδ2^+^ γδ T cells. However, γδ T cell clones produced lower levels of IFNγ (*p* = 0.001) and higher levels of IL-4 than αβ T cell clones (*p* < 0.02) [42]. In addition, in one patient with RA, LGL γδ cells, that expressed a Vγ9^-^Vδ2^+^ phenotype, and constituted 60% of the PB T cells, did not proliferate, but did secrete TNFα when triggered with anti-CD3, and the addition of these cells to decreased their secretion of immunoglobulin (Ig) M from pokeweed mitogen-stimulated B cells from the patient, while augmenting IgG secretion [43]. In contrast to these potential pro-inflammatory and immunogenic functions, CD4^+^ Th17 cells but not γδ T cells, were found in apposition to tartrate-resistant acid phosphatase positive osteoclasts in subchondral areas of inflamed joints in mice with collagen induced arthritis (CIA), and this pattern was reproduced in synovial biopsies of patients with RA [44]. Thus, γδ T cells of RA patients exhibit functional properties including antigen presentation, help for antibody production and predominantly TH1 like cytokine profiles, but may play a less significant role in bone resorption during the inflammatory process.

### 2.5. Responses to Putative Antigen

In a pioneering study, clones from rheumatoid synovium expressing Vγ9δ2 were shown to respond to synthetic alkyl phosphates and particularly to monoethyl phosphate (MEP) [45]. Later on, γδ T cells from synovial fluid were also shown to be reactive to isopentenyl pyrophosphate (IPP), an important endogenously produced phosphoantigen and an additive effect of a low dose of ethanol increased IPP induced proliferation of synovial γδ T cells, with similar findings applying to normal PB γδ T cells [46]. Other antigens for RA derived γδ T cells have been reported sporadically. Thus, among 15 T cell clones reactive with aggrecan, (a proteoglycan that binds to glycosaminoglycan hyaluronan through its amino terminal, the G1 globular domain), that were derived from PB of RA patients, 2 were γδ TCR^+^ [47]. In addition, six human Vδ1^+^ T cell lines were derived from RA SF, and were shown to selectively lyse Daudi, but not K562 cells in an MHC-unrestricted manner which was inhibited by anti-CD3 mAb. Since cold target inhibition assays showed that cytotoxicity was competitively inhibited by autologous and allogeneic primarily cultured RA synovial cells as well as synovial sarcoma and chondrosarcoma lines whereas PB did not inhibit this cytotoxicity, the authors concluded that Vδ1^+^ T cells in rheumatoid synovial may recognize an antigen which is commonly expressed on cells derived from RA synovium [48]. A further antigenic reactivity was suggested by the proliferative response of mononuclear cells (MC) in SF and PB of patients with to mycobacterial 65 kDa heat shock protein (HSP65). Higher response of SFMC than PBMC to HSP65 was noted in 14 of 19 patients with RA, and stimulation indexes of RA-SFMC correlated significantly with HLA-DR^+^ γδ^+^ T cell percentage suggesting reactivity of SFMC γδ T cells with HSP65 [49]. Finally, in one study, RA-derived Vγ9Vδ2 T cell clones, appeared to display dual antigenic recognition: a nonclonal, MHC-unrestricted recognition of mycobacteria, and a clonal recognition of a short tetanus toxin peptide presented by HLA-DRw53, a nonpolymorphic class II MHC molecule associated with susceptibility to rheumatoid arthritis. This was the first evidence for dual antigenic recognition by γδ T cells and suggested that Vγ9Vδ2 T cells could recognize nominal antigenic peptides presented by class II MHC molecules [50]. Taken together, these reports support the idea that Vδ1^+^ γδ T cells in the synovium may be responding to local antigens expressed on synoviocytes, whereas the Vγ9Vδ2 T cells that enter the joint in response to inflammatory stimuli, participate locally by recognizing phosphoantigens presented by butyrophilins expressed in the synovium (Figure 1).

### 2.6. Relationship to Disease Activity and Severity

Among 50 RA patients, γδ T cells in PB significantly decreased in negative correlation with the value of C-reactive protein (CRP), a marker of systemic inflammation, although they had no correlation with the titer of RF [22]. Furthermore, in patients with RA-associated neutropenia and expansions of LGL γδ T cells in the PB, absence of Vδ2^+^ T cells along with prominent Vδ1^+^ T cells in PB was associated with longstanding severe disease whereas patients with proliferation of Vδ2 cells had less severe disease, suggesting an association of the composition of the peripheral γδ T cell repertoire and disease severity [51]. In this regard, it was further shown that among 19 adults with early active RA, 80% of whom were RF^+^ or CCP^+^ and on no current steroid treatment, treatment with 99Tc-methylene diphosphonate (99Tc-MDP) increased Vγ9δ2 cells and Tregs (which were lower before treatment). Disease activity score (DAS)28, that evaluates disease activity in RA, decreased, as did concentrations of serum TNFα, IL-10, IL-6 during treatment with 99Tc-MDP, whereas IFNγ was unchanged and TGFβ increased [26]. Similarly, among sixty-eight patients with RA, % of Vδ2 T cells in PB correlated inversely with erythrocyte sedimentation rate (ESR), CRP, and DAS28 [27]. In another study, it was shown that among early RA patients of whom 8/10 were ACPA^+^, there was an increase of Vγ δ^+^CD27-CD45RA^+^ TEMRA cells, which correlated with DAS28. These cells produced high levels of IL-6, 8, and IFNγ ex-vivo, as well as in vitro, after stimulation with phorbol myristate acetate (PMA) and ionomycin. Furthermore, the number of cytokine producing cells correlated with DAS28 [25]. In all, it appears that the levels of γδ T cells, in particular that of the Vγ9Vδ2^+^ subset correlates with disease activity, decreasing with active disease while acquiring a TEMRA phenotype, and increasing during remission after treatment. A summary of how γδ T cells participate in the immune process of RA based on the data described in the preceding paragraphs, is presented in Table 1 and in graphic form in Figure 1.

## 3. Rodent Models of Rheumatoid Arthritis

### 3.1. Rat Models

In Mycobacterium tuberculosis-induced rat adjuvant arthritis (AA), protocols to deplete of TCR γδ (bright) cells in PB and lymph nodes, did not influence clinical parameters. If rats were treated before the clinical peak of adjuvant arthritis, however, joint destruction was significantly more severe than in vehicle-treated rats. The critical time window of intervention seemed to be limited to the span between the onset and the clinical peak of synovitis, since only anti γδ TCR treatment given in this phase, and not protocols administered before induction or around the peak of the disease, aggravated the degree of joint destruction [52]. In another study of AA in rats mediated by T lymphocytes specific for Mycobacterium tuberculosis, T cells bearing the αβ TCR were depleted from circulation by treatment with a mAb against the rat αβ TCR which efficiently suppressed existing disease. By contrast, there was no evidence that γδ T cells contributed to AA induction [53]. Likewise, in oil-induced arthritis, a genetically restricted polyarthritis that develops in the DA rats after injection of the mineral oil Freund’s incomplete adjuvant, disease was suppressed by CD8^+^ T cells but not by depletion of γδ T cells with a mAb (mAb) [54]. Finally, in the model of intradermal injection of squalene, a role for genes within the major histocompatibility complex, was concluded from comparative studies of MHC congenic rat strains. Treatment with anti αβ TCR but not anti γδ TCR prevented disease [55]. In conclusion of these studies, it appeared that in these models of rat arthritis, γδ T cells may play a role if any, during the effector rather than induction phases of the disease. 

### 3.2. Murine Model

In collagen induce arthritis (CIA)—induced by injections of collagen II in complete Freund adjuvant (CFA) with mycobacterium butyricum—in B10.Q male and DBA/1 female mice, CIA was no different in TCRδ^−/−^ mice than in controls, but was, in contrast, abrogated in TCRβ^−/−^ mice. The authors concluded that αβ T cells are necessary for CIA development and for an IgG response towards CII, whereas γδ T cells are neither necessary nor sufficient for development of CIA [56]. In another study, moreover, a mAb to TCR γδ had no effect, and actually slightly worsened arthritis, despite the fact that γδ T cells consisted up to 35% of the total T cells in the joints of mice with CIA. Some γδ T cells using Vγ1, -2, -4, and -6 and Vδ1, -2, -5, and -7 were found in the joints of normal mice, and this repertoire was similar to that found in arthritis joints [57]. These results therefore recapitulated those found in rat arthritis as detailed above.

However, when the individual responses of the two mains peripheral γδ T cell subsets, Vγ1^+^ and Vγ4^+^ cells, during CIA was examined, a more complex scenario unfolded. Thus, whereas both subsets increased in number, only the Vγ4^+^ cells became activated during CIA. These Vγ4^+^ cells appeared to be antigen (Ag)-selected, based on preferential Vγ4/Vδ4 pairing and very limited TCR junctions. Furthermore, in both the draining lymph node and the joints, the vast majority of the Vγ4/Vδ4^+^ cells produced IL-17, a key cytokine in the development of CIA. In fact, the number of IL-17-producing Vγ4^+^γδ T cells in the draining lymph nodes was found to be equivalent to the number of CD4^+^αβ TCR Th-17 cells. When mice were depleted of Vγ4^+^ cells, clinical disease scores were significantly reduced and the incidence of disease was lowered. A decrease in total IgG and IgG2a anti-collagen antibodies (Abs) was also seen. These results suggested that Vγ4Vδ4^+^ γδ T cells exacerbate CIA through their production of IL-17 [58]. 

Further support for the role of γδ T cells in antigen induced arthritis, was obtained in a model wherein methylated Bovine Serum Albumin (mBSA, 8 mg/mL) was emulsified in an equal volume of CFA containing heat-killed M. tuberculosis. At day 7, mice were immunized intradermally with mBSA/CFA and a week later, mBSA was injected intra-articularly to induce mono-arthritis. Inflammation in the joint was associated with high levels of IL-17 producing γδ T cells, and the expression of retinoic acid receptor related orphan receptor gamma (RORγ)t was dependent upon IL-23 suggesting that IL-23 regulates IL-17A and RORγt expression in γδ T cells in arthritis [59]. Furthermore, in both CIA and as well as in samples from patients with RA, an inhibitor of RORγT suppressed IL-17 production in γδ T cells stimulated with IL1β and IL23 ^+^ IPP [60]. Further mechanistic research came from a study to understand the role of IFN-lambda1 (IL-29), the main cytokine of class II cytokines (including IL-10 and IFNα β) in humans. This cytokine is not expressed in mice, where IL-28A/B instead, plays the major role. In the setting of CIA in male DBA/1OlaHsd mice and IL28-/- mice, therapeutic administration of IL-28A decreased IL-1β, IL23, and Th17 and γδ T cells producing IL-17 in the draining lymph nodes but not in PB. The target of IL-28 was neutrophils, due to their expression of IL-28R, suggesting a neutrophil mediated mechanism for γδ T cell involvement in CIA [61]. Another model studied the role of ES-62, a phosphorylcholine (PC)-containing glycoprotein secreted by the filarial nematode Acanthocheilonema viteae that acts to modulate the host immune response in order to promote the establishment of chronic helminth infection. ES-62 selectively induced toll like receptor (TLR) 4^+^ γδ T cells with the capacity to produce IL-22 but not IL-17 during CIA [62]. In addition, ES-62 down-regulated IL-17 responses in mice with CIA by targeting a complex IL-17-producing network, involving signaling between dendritic cells and γδ or CD4^+^ T cells. Thus, although it did not inhibit IL-17 produced by direct activation with IL-1/IL-23, ES-62 modulated the migration of γδ T cells by direct suppression of CD44 up-regulation and, as evidenced by in situ analysis, dramatically reduced levels of IL-17-producing cells, including lymphocytes, infiltrating the joint [63]. In yet another model, IL-10 receptor dominant-negative transgenic (Tg) and control mice were immunized with bovine type II collagen to induce arthritis. Blocking IL-10 signaling in T cells rendered mice highly susceptible to CIA. The suppressive function of CD4^+^ Foxp3^+^ regulatory T cells was significantly impaired in Tg mice because of the reduced ability of Tregs from Tg mice to maintain their levels of Foxp3. The higher level of IL-17 mRNA detected in inflammatory joints of Tg mice, was attributed to the recruitment of IL-17^+^ γδ T cells into the arthritic joints since IL-10 deficiency did not affect the percent of CD4^+^ IL-17- producing cells in the joint [64]. 

γδ T cells were found to be the predominant population among IL-17-producing cells in the swollen joints of mice with CIA, and the absolute numbers of these cells increased in parallel with disease activity. However, IL-17-producing γδ T cells expressed chemokine receptor 6 were maintained by IL-23 but not by type II collagen in vitro, and were induced antigen independently in vivo. Furthermore, IL-17 production by γδ T cells was induced by IL-1β plus IL-23 independently of TCR triggering. However, in autoimmune arthritis in SKG mice which is induced using zymosan as an adjuvant, in contrast to what was observed in mice with CIA, IL-17-producing γδ T cells were nearly absent in the affected joints. In this study, it was noted in addition, that in joints of patients with RA, IL-17-producing γδ T cells were rarely observed, whereas Th1 cells were predominant [65]. Likewise, as previously noted, it has been found that in CIA, CD4^+^ Th17, and IL-17 producing γδ T cells in the joints of arthritic mice similarly induced osteoclastogenesis in vitro. However, individual depletion and adoptive transfer studies revealed that in vivo, Th17 cells dominated with regard to bone destruction. Thus, unlike γδ T cells, Th17 cells were found in apposition to tartrate-resistant acid phosphatase positive osteoclasts in subchondral areas of inflamed joints, a pattern reproduced in patient biopsies [44]. 

Taken together, while clearly demonstrating involvement of γδ T cells in experimental arthritis, these data highlight the need to dissect subsets of γδ T cells when analyzing their role in pathogenesis of antigen induced arthritis in mice, while supporting the idea that their role is not directly associated with a direct response to the instigating antigen, but rather is related to effector mechanisms such as IL-23 induced production of IL-17 at the site of inflammation. 

Indeed, several models support the idea that in classical antigen induced arthritis, γδ T cells play an effector role downstream of and independent of direct antigen recognition. For example, it was demonstrated that, in the induction phase of CIA, CD4^+^Th17 cells in the lamina propria are activated. In CD4^+^Cre RORγ floxed mice arthritis was mitigated, despite continued production of IL-17 by γδ T cells [66]. By contrast, in arthritis that does not require any antigen to induce disease, but rather is induced by injected gene transfer of IL-23 in B10.RIII mice, γδ T cell depletion with mAb decreased neutrophils in joints and spleen while increasing IL-27 production by neutrophils and activated macrophages, resulting in abrogation of the arthritis. Blocking with anti TCR γδ mAb also resulted in reduced IL-17 but not TNFα, interferon (IFN)γ or interleukin (IL)-6. Thus, in this non antigen requiring form of arthritis, γδ T cells played a major role. It was further shown that IL-27 itself inhibited γδ T cells and reduced IL-23 induced arthritis [67]. In addition, IL-1 receptor(R) antagonist (a)-deficient (Il1rn^-/-^) mice spontaneously develop arthritis in an IL-17- and T-cell dependent manner suggesting that excess IL-1 signaling caused by IL-1Ra deficiency induces IL-17 production from T cells and the development of arthritis. IL-1R and IL-23R expressing Vγ6^+^ γδ IL 17 cells expressing high levels of C-C chemokine receptor (CCR2) type 2 are the main producers of IL-17 in joints of Il1rn ^-/-^ mice. Importantly, without CD4 cells, no arthritis occurred, and the CD4^+^ T cells were responsible for inducing C-C motif chemokine ligand (CCL) 2 in the joints, that attracted the pathogenic γδ T cells [68]. Interestingly, in yet another model it was shown that pathogenic γδ T cells may be under the control of other subsets of T cells. Thus, mice were given Salmonella enterica serovar Enteritidis #5694 by gavage. BALB/c Jα18−/− mice KO mice and anti CD1d treated mice developed more severe intestinal inflammation and worse arthritis. Infected mice had a higher percentage of IL-17 producing γδ T cells and depletion with anti γδ TCR partially abrogated joint inflammation. Mice treated with α galcer to activate induced natural killer (iNKT) T cells had less IL-17 γδ T cells and less arthritis but an increase of Th17 cells suggesting the pivotal role of IL-17 producing γδ T cells in this model [69]. Finally, a single intraperitoneal injection of mannan from the yeast *S. cerevisiae* into B10Q. Ncf1m1j/m1j (reactive oxygen species (ROS) deficient) mice resulted in a worse arthritis and psoriasis than that developing in wildtype B10Q mice. Arthritis was mediated by IL-17, and in this model the source of the cytokine was γδ and not αβ T cells. The secretion of IL-17 was dependent on TNFα produced by macrophages. It was thought that TLR2 expression by macrophages and γδ T cells might be responsible for the effect of mannan, revealing a mechanism for activation of pathogenic γδ T cells independent of a nominal antigen [70]. 

The main findings of these experimental models are summarized in Table 2. The cumulative data suggest that specific subsets of γδ T cells play an important role in the inflammatory response in the joint space in models of arthritis, primarily by secreting IL-17. Furthermore, this response appears to be independent of the inciting protein auto-antigen (e.g., collagen) used to induce disease. Thus, it appears that γδ T cell responses in arthritis are dependent, upon non TCR driven mechanisms, including cytokines (IL-1, IL-23, and IL-28) and chemokines affecting homing to the synovium, although a specific contribution of certain antigen selected γδ T cells (e.g., Vγ4Vδ4^+^ T cells) may also play a role. 

## 4. Juvenile Idiopathic Arthritis

### 4.1. Numerical Evaluation and Relationship to Disease Activity

CD4^-^CD8^-^ double negative (DN) and γδ T cell levels were measured in 42 patients with active JIA and in 10 HCs who were comparable for age by an immunofluorescence double staining procedure. All 3 JIA onset types had DN and γδ T cell levels not significantly different from those of controls. No correlation was found between DN or γδ T cell levels and ESR values or the number of active joints. When patients were divided according to treatment, DN and γδ T cell levels were significantly lower (*p* = 0.001, *p* = 0.02, respectively) in patients receiving methotrexate (MTX) than in patients not receiving MTX [71]. However, in another study of JIA, 80% of CD4^-^CD8^-^ of T cells were γδ T cells and γδ cells secreting IL-17 were positively correlated with innate lymphoid cells (ILC) type 3 secreting IL-17, which in turn correlated with disease severity by physician visual analogue score (VAS) [72]. In another study, when analyzed in relation to subsets of JIA, an elevated percentage of γδ T cells in PB was found in quiescent systemic JIA which decreased in active disease [73].

In studies of γδ T cells in the SF, it was found that in oligoarticular JIA, SF contained higher Treg (10.01% vs. 2.66% of CD3^+^CD4^+^) and γδ T cells (20.29% vs. 10.58% of CD3) than PB, and that higher levels of γδ T cells 919% vs. 15%) predicted more relapse within a median of 35.6 months [74]. Moreover, in another study of the SF of children with JRA, there were significantly more Vδ1^+^CD69^+^ and Vδ2^+^CD69^+^ cells compared with the PB of the same patients. The majority of the Vδ1^+^ cells expressed the high molecular weight isoform (CD45RA^+^) while most of the Vδ2^+^ cells carried the low molecular weight variant (CD45RO^+^) of this molecule [75]. In addition, when samples of PB (*n* = 25) and SF (*n* = 93) were analyzed by flow cytometry in 93 JIA patients, Vδ1^+^ and Vγ9^+^ γδ T cell subsets were detected in SF of all patients. However, only the percentage of Vδ1^+^ cells was higher in SF compared to PB (*p* < 0.01). Interestingly, although the distribution of γδ T cell subsets was similar in different JIA subgroups, antinuclear antibody (ANA)-positive patients had a higher percentage of SF Vδ1^+^ T cells than ANA-negative patients (*p* < 0.01). The percentage of SF Vδ1^+^ T cells was inversely associated with age at onset, recurrence of synovitis, and ESR; and that of SF Vγ9^+^ T cells was inversely correlated with age at onset and was higher in patients who recovered from disease (*n* = 15). IPP-induced expansion of SF Vγ9^+^ T cells correlated with disease remission, whereas the expansion of SF Vγ9^+^ T cells in media with IL-2 alone was significantly greater in patients with uveitis [76]. Taken together, these reports suggest a critical role for γδ T cells and their subsets in JIA, which may fluctuate with disease subtypes. 

### 4.2. Functional Characteristics

In addition to IL-6 and IL-18, IL-17A was prevalent in sera from patients with active systemic JIA, while ex vivo (inactive disease) and in vitro experiments revealed that γδ T cells overexpressed this cytokine. This was not seen with CD4^+^ T cells, which expressed strikingly low levels of IFNγ. Therapeutic IL-1 blockade was associated with partial normalization of cytokine expression phenotypes. Culturing healthy donor γδ T cells in serum from systemic JIA patients or in medium spiked with IL-1β, IL-18, and S100A12 induced IL-17 overexpression at levels similar to those observed in the patients’ cells and anti IL-1 blocked the increase [77]. Another study of JIA with or without enthesitis revealed an increase of γδ T cells in JIA patients (9.3 ± 4.6% vs. 5.12 ± 2.61%) associated with an increase of IL17^+^γδ^+^ and decrease of IFNγ^+^ γδ cells relative to controls [19]. Finally, among JIA patients, SF Vγ9^+^ T cells, expressed higher CD69 than those in PB. Moreover, only SF Vγ9^+^ T cells were further activated by contact with synovial fibroblasts to express more CD69 and secrete TNFα and these functions were augmented by zoledronate, which also augmented IFNγ by SF Vγ9^+^ T cells to a greater degree than PB Vγ9^+^ T cells. Proliferation of these cells was suppressed by contact with synovial fluid regulatory T cells. Furthermore, cell-cell contact with the Vγ9 T cells induced synovial fibroblast apoptosis [78]. Together these results suggest complex pathogenic as well as possible protective roles for subsets of γδ T cells in JIA.

## 5. Murine Models Relevant to JIA

To study a model of systemic JIA, wild-type (WT) and IFNγ-knockout (KO) BALB/c mice were challenged with CFA, and clinical symptoms and biologic characteristics analyzed. In WT mice, CFA injection resulted in splenomegaly, lymphadenopathy, neutrophilia, thrombocytosis, and increased cytokine expression. In the absence of IFNγ, arthritis, anemia, hemophagocytosis, abundance of immature blood cells, and increased levels of IL-6, all of which are reminiscent of the symptoms of systemic JIA, were noted. CFA-challenged IFN-KO mice showed increased expression of IL-17 by CD4 T cells and by innate γδ T cells. Inflammatory and hematologic changes were prevented by treatment with mAb to IL-12/IL-23p40 and by anti IL-17 antibodies, indicating a role of IL-17- producing γδ T cells in this syndrome [79] 

## 6. Ankylosing Spondylitis (AS)

In a pioneering study, IL-23 receptor (IL-23R) expression in T cells was determined in 17 active AS, 8 patients with PsA, 9 patients with RA and 20 HC. The proportion of IL-23R–expressing T cells in the periphery was 2-fold higher in AS patients than in HC, and this was specifically driven by a 3-fold increase in IL-23R^+^ γδ T cells in AS patients. Increased IL23R expression on γδ T cells was also associated with enhanced IL-17 secretion, with no observable IL-17 production from IL-23R- γδ T cells in AS patients. Furthermore, γδ T cells from AS patients were heavily skewed toward IL-17 production in response to stimulation with IL-23 and/or anti-CD3/CD28. In PsA patients, IL23R^+^ γδ T cells were also higher but they were not IL-17 secretors. Interestingly, IL23R^+^ γδ T cells were also found in joints of patients with crystal induced arthritis [80]. In another study, RORγt^+^ Tbetlo PLZF− iNKT and γδ TCR “high” T cell subsets were found in healthy PB and in AS patients. RORγt^+^ iNKT and γδ-hi T cells showed IL-23 mediated Th17-like immune responses and were clearly enriched within inflamed joints of spondyloarthritis patients, where they acted as major IL-17 secretors. The RORγt γδ TCR high cells were mainly Vδ^+^, expressed IL-23R mRNA, and responded to IL-23 by producing IL-17 and IL-22. The frequency of γδ TCR “high” cells and also of IL23R^+^ γδ T cells was higher in SF than in PB, cells. RORγt inhibition selectively targeted IL-17 producing iNKT and γδ-T cells enriched in spondyloarthritis patients [81]. In addition, IL-17 and granulocyte monocyte colony stimulating factor (GM-CSF) double producing γδ T cells were found to be increased in PB of AS patients when compared to RA patients and normal PB, similar to what was found with respect to CD4, CD8 and natural-killer (NK) cells [82]. Another study confirmed that double expressing IL-17 and GMCSF γδ as well as CD4, CD8, and ILC cells are found in synovium and increased in PB of AS patients, whereas there was a decreased percent of γδ T cells in their PB relative to normal [83]. Interestingly, in contrast to IL23R^+^ IL-17^+^ Vδ1^+^ T cells that are expanded in AS, Vδ2^+^ γδ T cells were found to be increased in patients with AS who were receiving infliximab (anti TNFα) therapy, relative to those not receiving this drug and HC. Most of these Vδ2^+^ T cells secreted IFNγ and not IL-17 [84]. 

A single study addressed antigenic reactivities in AS. Four randomly derived synovial V γ9 ^+^ γδ-TCR^+^ clones from SF of patients with spondyloarthropathy killed both autologous and allogeneic target cells when infected with live Yersinia or Salmonella and also uninfected Daudi cells expressing GroEL heat shock protein. One clone was specific for Yersinia-infected targets. Three γδ-TCR^+^ clones were cytotoxic when uninfected autologous or allogeneic targets were employed. Polymorphic “classical” MHC class I or class II molecules were not used as restriction elements for cytotoxicity [85]. Taken together, the results of these studies support a unique and critical role of IL-17 secreting, IL-23 responsive Vδ1^+^ γδ T cells in the pathogenesis of AS.

### Murine Model Relevant to Ankylosing Spondylitis

In a pivotal study, γδ T cells were found to be abundant in uninflamed entheseal tissue and constituted the large majority of RORγt^+^ IL-23R^+^ enthesis resident lymphocytes. These cells were generated from fetal thymus dependent Vγ6^+^ CD44^+^ CD27^−^ γδ T cells and were the main source of IL-17A at the enthesis. Under inflammatory conditions supplied by introduction of the IL-23 gene, γδ T cells increased in number at the Achilles tendon enthesis, aortic root, and adjacent to the ciliary body, consisting 25% relative to 1% in the lymph node and 40% of the Vγ6^+^ cells secreted IL-17 [86].

## 7. Systemic Lupus Erythematosus

### 7.1. In Vivo Levels of γδ T Cells and Their Correlation with Disease Activity

In a study of 21 patients with SLE and 17 healthy donors, γδ T cells were significantly decreased whereas the frequencies of HLA-DR^+^ γδ T cells and CD69^+^ γδ T cells were significantly higher in patients (*p* < 0.001 and *p* < 0.05 respectively) compared with HC. CD94^+^ γδ T cells and CD94^+^ NKG2A^+^γδ T cells were, in contrast, lower in patients with SLE compared with HC (*p* < 0.001) [87]. In another study, PB-derived γδ T cells were isolated from 14 healthy volunteers and 22 SLE patients before and after 4 and 12 weeks following the onset of glucocorticoids (GC), mycophenolate mofetil (MMF), or hydroxychloroquine (HCQ) treatment. PB γδ T cells in general, and Vγ9^+^ γδ T cells and TNFα/IL-17-secreting CD4^-^CD8^−^ γδ T cell subsets, in particular, were decreased in SLE compared to HCs, but the numbers of the γδ T cell subsets reached levels similar to those of HCs in responders, but not in non responders, to therapy. There was an inverse correlation between SLEDAI scores and γδ T cell c ompartments, especially with respect to TNFα^+^ γδ T cells, TNFα^+^Vγ9^+^ γδ T cells and IL-17^+^ CD4^-^CD8^−^ γδ T cell subsets and complement component 3 levels positively correlated with IL-17 producing γδ T cells [88]. In a study addressing subsets of γδ T cells, a significant decrease in the proportions of total γδ T cells and the Vδ2 subset in new-onset SLE patients in comparison with HCs was found (*p* = 0.016 for γδ T cells, *p* = 0.003 for the Vδ2 subset, along with an increased Vδ1 proportion in inactive SLE patients (*p* = 0.004). The proportion of the CD27^+^CD45RA^-^subset (central memory γδ T cells), was significantly reduced in both active and inactive SLE patients compared with HC. Absolute γδ T cell counts were also significantly decreased in active SLE patients, as were the Vδ1 and Vδ2 subsets. Upon treatment, however, γδ T cell counts increased to normal levels with Vδ1 cell counts increasing to levels higher than those observed in healthy donors. Although Vδ2 cell counts rose to some extent, they remained significantly lower than in healthy donors. Furthermore, there was a negative correlation between the SLE Disease Activity Index (SLEDAI) and Vδ and Vδ2 γδ T cell counts. CD27^+^CD45RA^+^ γδ T cell numbers also negatively correlated with the SLEDAI. CD27^+^CD45RA^-^ Vδ cells displayed Foxp3, suggesting their regulatory potential and Foxp3 mRNA in γδ T cells was markedly augmented following transforming growth factor (TGF) β treatment in vitro. Furthermore, Foxp3^+^ γδ T subsets suppressed CD4^+^ T cells, suggesting these cells could modulate disease activity via suppression of pathogenic CD4^+^ T cells [89]. In contrast to PB, the percentage of γδ T lymphocytes in the skin of SLE patients, was twice higher (22.0 ± 9.5%) than in the skin of healthy persons (11.1 ± 5.5%) (*p* < 0.002) and a higher percentage was noted in patients with active disease (27.0 ± 9.4%) than in non-active SLE (16.6 ± 5.6%) (*p* < 0.002). Similar differences were noted in the percentage of Vδ2^+^ and Vγ9^+^ subpopulations in SLE patients when compared to healthy persons. With respect to the Vδ3 subset, a higher percentage was observed in patients with active SLE (10.5 ± 4.8%) than in patients with non-active SLE (6.8 ± 3.5%; *p* < 0.03) and in healthy persons (3.6 ± 3.1%, *p* < 0.02). By contrast, CD3^+^ lymphocytes in the skin of SLE patients and healthy persons was, however similar (81.4 ± 7.9 and 83.0 ± 13.4%, respectively; *p* > 0.05). Furthermore, there was a positive correlation between the percentage of γδ T lymphocytes in the skin and the activity of the disease (r = 0.594, *p* < 0.001). Confirming previous studies, there was a significantly lower number of γδ T cells in PB of SLE patients (26.4 ± 16.9/mL) than in healthy persons (55.3 ± 20.6/mL) (*p* < 0.001) but no statistically significant correlation between the concentration of these cells and clinical activity of the disease was found [90]. Thus, γδ T cells and their subsets are depleted in the PB of SLE patients, in inverse correlation with measures of disease activity (Table 1).

### 7.2. In Vitro Studies of γδ T Cells from SLE Patients

T cell lines were derived from a patient with subacute cutaneous lupus after treatment with intravenous pulse cyclophosphamide by selection of mitotically active, hypoxanthine-guanine phospho-ribosyltransferase-deficient (HPRT-D) T cells in a medium containing 6-thioguanine. When HPRT-D cell lines were derived 6 days after pulse cyclophosphamide (CYC) treatment, they were predominantly CD8^+^ and TCR γδ^+^, and produced IFNγ. Cell lines derived 21 days after CYC treatment were in contrast CD4^+^, TCRαβ^+^ and produced both IFNγ and IL-4, suggesting involvement of mitotically active γδ and αβ T cells at different phases of the disease [96]. Further evidence for the relevance of γδ T cells came from a study of a total of 396 IL-2-dependent T-cell lines from the in vivo activated T cells of 5 patients with lupus nephritis. Only 59 (~15%) selectively augmented the production of pathogenic anti-DNA autoantibodies that were IgG in class, cationic in charge, specific for native DNA, and clonally restricted in spectratype. Forty-nine of the autoantibody-inducing Th lines were CD4^+^ and expressed the αβ TCR. The other 10 were CD4^−^8^−^, of which 3 expressed the αβ TCR and 7—the γδ TCR. The autoreactive responses of the CD4^+^ Th lines were restricted to HLA class II antigens, whereas endogenous heat shock or stress proteins of the HSP60 family that were expressed by the lupus patients’ B cells were involved in stimulating an autoreactive proliferation of the γδ Th cells. These autoantibody-inducing CD4^−^8^−^ γδ Th lines proliferated in response to HLA-mismatched antigen presenting cells from lupus patients but not normal subjects, and could also provide HLA-nonrestricted help for production of IgG anti-dsDNA autoantibodies by B cells from unrelated lupus patients. The autoreactive response of the γδ Th lines but not the αβ Th lines, could be blocked partially or completely by mAbs to the 65-kDa HSP whereas anti-HSP70 mAb had no significant effect. These studies demonstrated helper activity of certain γδ T cells in SLE [92]. From these studies, it is clear that γδ T cells of SLE patients harbor the potential to affect disease by encouraging the production of pathogenic anti DNA antibodies by patient B cells.

## 8. Murine Models

### 8.1. MRL/lpr Model

The congenic MRL/Mp-lpr/lpr strain develops a rapid form of MRL lupus, including the accumulation of CD4^−^CD8^−^B220^+^ T cells, due to a defect in the Fas apoptosis gene. Eight groups of homozygotic MRL/lpr mice, including TCRβ^+/+^ or TCRβ^−/−^, TCRδ^+/+^ or TCRδ^−/−^, and Fas^+/+^ or Ipr/lpr were studied. γδ T cell-deficient animals developed exacerbated disease, as evaluated by hyperglobulinemia, autoantibody titers, renal disease, and mortality, demonstrating a role for γδ T cells in the regulation of systemic autoimmunity. Thus, TCRδ^-^ lpr mice consistently developed accelerated and more severe disease than TCRβ^+^δ^+^ animals, as typified by glomerulonephritis, interstitial infiltrates, and perivasculitis, an effect more apparent in younger mice. Conventional CD4^+^ T cells were polyclonally expanded in these animals at the expense of classical lpr CD4^−^CD8^−^B220^+^ T cells, suggesting that γδ T cells regulate autoimmune responses by altering T cell development. γδ T cell-deficient animals of both Fas^+^ and lpr genotypes develop higher intensity as well as higher titer autoantibodies than their TCRβδ^+^ counterparts. On the other hand, whereas TCRβ^-^γ^-^ sera contained no detectable antinuclear antibodies, TCRβ^−^ lpr mice, developed an incompletely penetrant autoimmune syndrome, suggesting that γδ T cells themselves were responsible for the development of elevated autoantibody titers in these animals. While mice possessing either αβ or γδ T cells were capable of producing substantial IL-2, IL-4, IFNy, and IL-10, TCR β−δ^-^ mice failed to produce IL-2 or IL-4 and generated greatly reduced levels of IFNγ and IL-10. Thus, γδ T cells in this autoimmune model of SLE have the capacity to provide T-dependent help and or regulation via cytokine production [97]. In another study of MRL/lpr mice CD4^+^B220^+^ cells accumulated during aging of mice and development of SLE, when they consist up to 5% of splenocytes. Of these, about 32% were found to express γδ TCR and produce IL-17. Furthermore, CD4^+^B220^+^ producing cells could enhance the proliferation of conventional CD4^+^B220^-^ pathogenic cells [98]. Mechanisms involved in the effects of γδ T cells in the MRL/lpr model were further explored in the Blk±.lpr mouse, in which Blk expression levels are reduced to levels comparable to those in individuals carrying a risk allele for BLK, which encodes B lymphoid kinase, a susceptibility gene for SLE, resulting in reduced gene expression. BLK was found to be expressed in IL17^+^ γδ T cells, as well as B cells and RORγ^+^ DNαβ cells and pDC. At 5 months of age, 60% of the Blk^+/−^.lpr mice, but none of the B6.lpr mice, displayed proteinuria. While there was no mesangial proliferation or any other signs of immune complex (IC)-induced inflammation or difference in the serum levels of ANAs between 5-month-old B6.lpr and Blk ^+/−^.lpr mice, there was narrowing of the capillary lumens and hyaline deposits were found in the glomeruli of Blk ^+/−^.lpr mice but not in those of B6.lpr mice, as was damage to the podocyte, a component of the glomerular filtration barrier. Thus, 5-month-old Blk^+/−^.lpr mice suffered from nephrosis, a kidney disease that is frequently observed in SLE patients with renal involvement due to lupus podocytopathy. In the BLK^+/−^.lpr mice there were more IL-17 and IFNγ secreting γδ T cells. These findings suggest that BLK risk alleles confer susceptibility to SLE through the dysregulation of a proinflammatory cytokine network including γδ T cells [93]. Another mechanism of γδ T cell involvement might be attributed to γδ T cell mediated apoptosis. Thus, it has been shown that the phosphopeptide P140 issued from the spliceosomal U1-70K small nuclear ribonucleoprotein (snRNP) protein is recognized by lupus CD4^+^ T cells, transiently abolishes T cell reactivity to other spliceosomal peptides in P140-treated MRL/lpr mice and ameliorates their clinical features. P140 peptide binds the constitutively-expressed chaperone HSC70 protein and induces apoptosis of activated MRL/lpr CD4^+^ T cells. When nine-week-old MRL/lpr mice received two successive intraperitoneal and intravenous administrations of anti-pan TCR γδ mAb eight and three days before P140 treatment, P140 induced no PBL apoptosis contrasting to its affect in untreated mice. Thus, in vivo P140 induces PBL apoptosis, leading to amelioration of SLE in MRL/lpr mice via a mechanism involving γδ T cells [94]. From these studies, salient features of which are summarized in Table 2, it is apparent that in the MRL/lpr model of autoimmunity resembling human SLE, γδ play both a regulatory role dampening pathogenic effects of αβ T cells, perhaps via apoptotic mechanisms, but may also have, in situations in which conventional T cells are suppressed, a pathogenic role that is partly controlled by the BLK gene.

### 8.2. NZB/NZW Model

In this model, CD1d-dependent T cells were depleted in order to investigate the role of these cells in genetically lupus-prone NZB/NZW F1 (BWF1) mice, by injections of 50 million irradiated CD1d-transfected A20 cells. This resulted in 50–75% reduction in NK1.1^+^ T cells, including CD4^+^NK1.1^+^, NK1.1^+^CD122^+^ or NK1.1^+^CD62L− T cells and NK1.1^+^ TCR γδ ^+^ T cells. This depletion of CD1d-reactive T cells in preclinical BWF1 mice resulted in disease acceleration with a significant increase in proteinuria and mortality. In older BWF1 mice having advanced nephritis, however, depletion resulted in some disease improvement. Thus, similar to the findings in the MRL/lpr model, and suggested by some of the human functional studies, γδ T cells bearing receptors that recognize CD1d-lipids along with other types of CD1d dependent T cells may participate in pathogenic and protective roles in SLE [99].

### 8.3. Pristane Induced Model

In a study of 8-week-old C57BL/6 and TCRδ^−/−^ mice that received a single i.p. injection of 0.5mL of pristine, the TCRδ^−/−^ mice developed milder glomerulonephritis, consistent with their decreased serum levels of lupus-related autoantibodies, when compared with wild type mice. It was also found that injection of CFA, but not alum immunization induces a subpopulation of CXCR5-expressing γδ T cells in the draining lymph nodes. TCRγδ^+^ CXCR5^+^ cells presented antigens to, and induced CXCR5 on, CD4 T cells by releasing Wingless-related integration site (Wnt) ligands to initiate the T follicular helper (Tfh) cell program. Accordingly, TCRδ^−/−^ mice had impaired germinal center formation, inefficient Tfh cell differentiation, and reduced serum levels of chicken ovalbumin (OVA)-specific antibodies after CFA/OVA immunization, suggesting that modulation of SLE by γδ T cells is also mediated by their ability to control TfH cell differentiation [100]. In addition, in the same model of pristane-induced SLE, increased IL-17F and IL-17A expression was found in renal CD4^+^ T cells and γδ T cells. Moreover, in this chronic model SLE, IL-17F–deficient mice developed less severe disease than wild-type mice, with respect to survival and renal injury. IL-17F induced expression of the neutrophil-attracting chemokines CXCL1 and CXCL5 in kidney cells, suggesting a role for IL-17 producing γδ T cells and Th17 cells in recruitment of neutrophils involved in renal inflammation in SLE [101]. A summary of the mode of participation of γδ T cells in SLE, including data from human and murine studies is presented in Table 1 and Table 2 and, in graphic and tabular forms in Figure 2 and Table 2.

## 9. Systemic Sclerosis

### 9.1. γδ T Cells in PB and Tissue of SSc Patients

Patients whose serum contained anti Scl-70 antibodies and patients with shorter disease duration had reduced number of γδ T cells in PBMC compared with controls (*p* < 0.05 and *p* < 0.01, respectively). Absolute values of γδ T cells were lowest for patients with a disease duration of less than three years (14.2 ± 5. 9 vs. 37.5 ± 12.5) cells/mm^3^, those with anti-Scl-70 antibodies (23.9 ± 10.4 vs. 30.3 ± 11.5) cells/mm^3^ and those with diffuse disease (23.7 ± 7.2 vs. 40.3 ± 25.1 cells/mm^3^). There was no significant difference in the proportion or absolute number of CD56 cells or γδ T cells between patients receiving or not receiving steroid medications. The proportion of γδ T cells was also significantly lower in the patient group (1.61 ± 0.52% vs. control 2.61 ± 0.46% (*p* < 0.05). Again, patients with early disease or with anti-Scl-70 antibodies accounted for the reduction of the proportion of γδ T cells [91]. In another study, 50 patients with SSc had lower values (both percentage and absolute number) of NKT cells (*p* < 0.01 and *p* < 0.003, respectively) and of γδ T cells (*p* < 0.01 and *p* < 0.005, respectively) [102]. In yet another cohort of 12 patients with SSc, when compared to 16 healthy volunteer donors, it was found that the Vδ1^+^ γδ T cell subset had significantly enhanced expression of both HLA-DR (83% of total Vδ1^+^ cells) and CD49d (90% of total Vδ1^+^ cells) compared with the controls (20.5% and 60%, respectively). The percentage of total Vδ1^+^ γδ T cells was also enhanced (2.7 vs. 0.8 *p* < 0.005), whereas the percent of total γδ T cells was similar to controls. Percentages and absolute numbers CD16, CD8, CD45RO, CD25, HLA-DR, CD54, and CD11a on total γδ T cells did not differ significantly from controls whereas CD49d^+^ γδ T cells were significantly increased in the patients (2.3%) compared with controls (0.5%). In the skin, the absolute numbers of γδ T cells were found in striking amounts in perivascular areas, particularly in the early edematous phase of SSc (22.58 in patients and 0 in controls); the majority of γδ T cells were Vδ1^+^ (19 in patients and 0 in controls). Moreover, even in the advanced phase of SSc, Vδ1^+^ T cells were also increased compared with controls (3.5 versus 0) [95]. In another study, however, the percent of Vδ1^+^ γδ T cells was significantly elevated among the PB T cells in patients without radiographic evidence of interstitial lung disease (ILD) but not in those with ILD (*n* = 7). Vγ9^+^ T cells were equally and persistently represented in the PB of patients irrespective of pulmonary disease or cyclophosphamide treatment, at levels similar to HC [103]. Interestingly, supporting this finding, the proportion of CD161^+^ Vδ1^+^ γδ T cells in PBMCs from the HCs, RA patients, and PM/DM was significantly lower than in SSc patients and similar results were found with regard to the absolute number of CD161^+^ Vδ1^+^ γδ T cells in PBMCs. Furthermore, the proportion of CD161^+^ Vδ1^+^ γδ T cells was significantly higher in IP-negative SSc patients compared with IP-positive SSc patients and HCs. However, CD161^+^ Vδ1^+^ γδ T cells were not altered by the presence of IP in patients with RA and PM/DM suggesting a disease specific relationship [104]. In another study, although no significant difference in number and proportion of γδ T cells was observed in SSc patients compared to HC, geometric mean fluorescence intensity (GMFI) of CD16 and CD69 on γδ T cells was significantly increased in patients with diffuse cutaneous SSc (dcSSc) compared to HCs while CD62L expression was significantly decreased. Additionally, γδ T cell infiltrations were observed in SSc patients’ skin [105]. In yet another study, absolute numbers of γδ T-cell subsets were, like some of the other reports cited above, were found to be significantly decreased in SSc patients, which was thought to reflect their mobilization to the inflamed skin. In addition, patients with pulmonary fibrosis showed a biased TCR repertoire, with a selected expansion of effector Vγ9^+^ γδ T cells [106]. In summary, despite a general reduction of γδ T cells in the PB of patients with diffuse disease, these data reveal a unique expansion of activated Vδ1^+^ T cells in SSc, which may be more pronounced in the absence of lung involvement. Moreover, these cells appear to express a CD161^+^ phenotype (Table 1).

### 9.2. Functions and Subsets of γδ T Cells in SSc

In assessing cytokine production and cytotoxic activity of circulating γδ T lymphocytes obtained from SSc patients and to evaluate their potential role during this disorder, both the proportion and the absolute number of IFNγ producing γδ T cells (i.e., displaying a Th1 polarization) in SSc were found to be significantly higher than the proportion and the absolute number of IL-4 producing γδ- T cells in SSc, or the proportion and the absolute number of IFNγ -producing γδ T cells in HC (*p* < 0.05 for both groups). Furthermore, the cytotoxic activity of enriched γδ T cells was significantly increased in SSc patients compared with controls. The results concerning the Vδ1^+^ T cell subset paralleled those of total γδ T lymphocytes. In contrast, αβ T cells from SSc displayed greater Th2 cytokine production. All these findings were independent of both disease subset and clinical status. Thus, although SSc is generally considered a Th2 autoimmune disease, Th1 polarization of γδ T cells and an increase in their cytotoxic activity is observed in SSc, suggesting that γδ T cells could have a relatively autonomous role in the pathogenesis in this disease [95].

Further functional studies revealed, in a study of 16 SSc patients and 16 HCs, that ex vivo triggering of patient PB Vγ9^+^ T cells with IPP plus IL-2-induced dose-dependent expansion, resulted in secretion of TNFα, and contact-dependent apoptosis of co-cultured fibroblasts, at levels similar to Vγ9^+^ T cells of controls [103]. In another study, isolated γδ T cells were then co-cultured with fibroblasts, and mRNA expressions of proalpha1(I) collagen and proalpha2(I) collagen (COL1A2) of fibroblasts were analyzed by real time RT-PCR. COL1A2 mRNA expression was significantly higher in fibroblasts cocultured with γδ T cells from SSc than that from HCs in cell contact independent manner [105]. In yet another study, γδ T-cell subsets displayed a Th1-type cytokine responses. However, cytotoxic properties showed significant disease-associated and subset-specific changes. SSc patients exhibited increased percentages of CD27^+^ γδ T cells expressing granzyme (GZM) B or perforin and upregulated GZMA expression in diffuse cutaneous SSc. Conversely, Eomesodermin (EOMES) and NKG2D were downregulated in both SSc γδ T-cell subsets vs. normal controls. Interestingly, patients with pulmonary fibrosis showed a biased TCR repertoire, with expansion of effector Vγ9^+^ γδ T cells associated with increased frequency of cells expressing GZMB, but decreased IFNγ production. The authors concluded that there is an increased cytotoxic activity and thus enhanced pathogenic potential of CD27^+^ γδ T cells in SSc [106]. These findings are supported by an early study, in which γδ- and αβ-T cells were isolated from PB of SSc patients with early diffuse disease and of matched control subjects by an immunomagnetic method after stimulation with mycobacterium lysate and IL-2 for 2 weeks. It was found that SSc γδ-T cells adhered to cultured endothelial cells (EC) and proliferated at higher rates than control cells. Furthermore, significant EC cytotoxicity by SSc γδ was seen. The cytotoxicity was blocked by addition of anti-γδ-TCR antibody and by anti-granzyme A antibody but not by anti-MHC class I and II antibodies. Expression of granzyme A mRNA was seen in five/five SSc γδ-T cells and in one/five control cells. αβ-T cells from both SSc and control subjects were significantly less interactive with EC than γδ-T cells [107].

In another detailed study, 17 RA patients, 35 with SSc, 14 with polymyositis/dermatomyositis (PM/DM) and 22 disease-free HCs were analyzed. The proportion of CD161^+^ Vδ1^+^ γδ T cells in PBMCs correlated negatively with serum KL-6 values (indicative of active interstitial lung disease) among IP^+^ SSc patients, whereas that of total Vδ1^+^ γδ T cells and CD161-Vδ1^+^ γδ T cells did not. As indicated above, the proportion of CD161^+^ Vδ1^+^ γδ T cells in PBMCs was higher in SSc in particular in IP- patients (1.03% ± 0.30) compared with IP^+^ SSc patients. Intracellular staining of PMA^+^ionomycin triggered Vδ1^+^ cells from patients and controls revealed similar levels of IFNγ and other cytokines. CD161^+^ Vδ1^+^ T cells secreted more IFNγ but not IL-4, IL-17, or TNFα than CD161- Vδ1^+^ T cells in response PMA + ionomycin. However, in IP^+^ patients, CD161^+^ Vδ1^+^ cell lines produced less IFNγ than controls in response to triggering the TCR with anti Vδ1 mAb. Furthermore, CD161^+^ Vδ1^+^ γδ T cell lines of IP- SSc patients produced a significantly greater amount of CCL3 (which abrogates inhibiting effects of IFNγ on fibroblast proliferation) than HC CD161^+^ Vδ1^+^ γδ T cell lines, upon TCR stimulation [104]. Given the important suggested role of Vδ1^+^ γδ T cells with respect to pulmonary fibrosis, outlined in these studies, it is of interest that an increased percentage of γδ T cells expressing the TCR Vδ1 gene was detected in bronchoalveolar lavage fluid of patients with SSc. To estimate clonality of these Vδ1^+^ T cells, the diversity of Vδ1 junctional regions (V-D-J) was examined using RT-PCR to amplify TCR δ-chain transcripts isolated from PBMC, lung, esophagus, stomach, or skin of patients and controls. Limited diversity of Vδ1-Jδ junctional regions in SSc patients was demonstrated by comparing the size distribution of PCR-amplified junctional region cDNA from patients with that of controls. Sequence analyses confirmed that Vδ1-Jδ junctional regions from the blood of SSc patients had less diversity than those from controls, in that a significantly higher proportion of sequences were repeated in patients (54.4% vs. 19.4% in controls). Evidence for selection of the Vδ1^+^ T cells in the tissues of SSC patients came from the findings that the same Vδ1-J δ junctional sequences persisted in an individual patient over time and that identical junctional sequences were isolated from multiple sites. Analysis of deduced amino acid sequences revealed two clusters of similarities among the junctional regions from patients. These data suggest that expansion of Vδ1^+^ γδ T cells may be antigenically driven in SSc patients [108]. Further evidence supported the idea that SSc Vδ1^+^ T cells may recognize the autoantigen, cardiolipin in a CD1d dependent manner, and the outcome of this recognition is influenced by cross talk with activated Vγ9^+^ T cells. Thus, in vitro, cardiolipin (CL) decreased CD25 on Vγ9 T cells from SSc patients, an effect which was abrogated by zoledronate. Zoledronate, on the other hand, decreased CD25 expression in SSc Vδ1 T cells uniquely, whereas CL abrogated this decrease. Moreover, anti CD1d abrogated CL induced enhancement of CD25 expression of SSc Vδ1 cells. CL also decreased IFNγ secretion of SSc Vδ1^+^ T cells relative to HC. Thus, CD1d mediated interactions of Vδ1^+^ T cells in SSc may play a profibrotic role via effects on pro and anti fibrotic cytokines, in particular in the context of simultaneous Vγ9^+^ T cell triggering [109].

On the other hand, because of the functional cytokine producing and cytotoxic functions role of Vγ9^+^ T cells in SSc detailed above, it was of interest, that an SSc patient was described whose condition worsened dramatically after an infusion of zoledronate. In vitro, zoledronate increased TNFα secretion by patient Vγ9^+^ γδ T cells, and induced tissue factor-1 on monocytes, which could be abrogated by anti TNFα mAb. Zoledronate also induced IL-4 production in SSc Vδ1^+^ but not in HC Vδ1^+^ T cells. These findings suggest mechanisms wherein activating Vγ9^+^ γδ T cells in vivo could lead to deleterious effects in SSc [110]. On the other hand, when PBMC from SSc patients and HCs were stimulated by increasing concentrations of zoledronate, with or without IPP, and Vγ9^+^ T cell percentages were calculated using FACScan analysis it was found that higher concentrations of zoledronate were required for maximal proliferation of Vγ9^+^ T cells in 9 SSc patients compared to 9 HCs. When compared to stimulation by toxic shock syndrome toxin (TSST)-1, a non-Vγ9^+^ selective reagent that activates αβ T cells, secretion of the anti-fibrotic cytokines TNFα and IFNγ in response to the Vγ9 selective antigen IPP was relatively diminished in SSc compared to HC. In addition, reduction of procollagen secretion by fibroblasts cultured with supernatants of IPP-stimulated PBMC was observed in some SSc patients suggesting that although somewhat defective in their secretion of the anti fibrotic IFNγ, patient Vγ9^+^ T cells may maintain some, although insufficient, anti fibrotic characteristics [111]. Interestingly in this regard 18 patients with SSc received a single intravenous dose of 60 mg of pamidronate and were followed for 6 months to assess effects on cytokine production by PBMC. Unstimulated PBMC produced increased amounts of IFNγ and TNFα and reduced levels of TGFβ for up to 24 weeks after the infusion [112]. Taken together, these studies indicate important functional roles for γδ T cells in SSC patients. These cells, in particular the Vγ9^+^Vδ2^+^ subset, appear to be enriched for cytotoxic potential relative to HC, and when activated which may manifest anti endothelial and anti fibroblastic potential, while simultaneously producing pro-coagulant inducing (TNFα) and anti fibrotic (IFNγ) cytokines. In addition, the Vδ1 subset, cells of which may be recognizing lipid antigens presented by CD1d, appears to be distributed oligoclonally in tissues, where they may act in a deleterious profibrotic capacity mediated in part by profibrotic factors such as IL-4, CCL3, while producing lower level of the anti fibrotic IFNγ than healthy Vδ1^+^ T cells. Figure 3 shows, in graphic form, a model incorporating the above described modes of participation of γδ T cells in SSc. A review of prevalent animal models of SSc the reader is available in reference [113].

## 10. Concluding Remarks

Interestingly, the lead authors of the first papers to describe γδ TCR and γδ T cells were both rheumatologists, a profession which focuses on ARDs [4,5]. It is therefore gratifying to note from the comprehensive data presented above, that while γδ T cells are a highly conserved subset of T cells of basic importance to all aspects of immunobiology, the role of these cells in ARDs, in particular, cannot be ignored. Thus, we have shown that ARDs are associated with quantitative and qualitative perturbations of the systemic and localized distribution of specific γδ TCR and of the functional programs of the cells expressing these receptors. These, in turn, may have profound effects on the development, manifestations, and outcome of ARDs. Animal models suggest that γδ T cells exert their influence via modulatory effects on classical auto-antigen reactive αβ T cells and B cells—central mediators of ARDs. However, they also playing an independent effector pro-inflammatory role, mediated by their innate ability to secrete IL-17, TNFα, and IFNγ in a non- antigen driven fashion. Looming above all this is a crucial, largely unanswered central question: what are the specific auto-antigens for γδ TCR in ARDs? Likewise, what is the role of and how are putative antigen presenting molecules such as CD1d, MR1, and butyrophilins involved? Future studies will likely address these questions and the answers will likely lead to a novel and deeper understanding of ARDs and ways to treat them, for the benefit of patients suffering from these chronic diseases.

## Figures and Tables

**Figure 1 cells-09-00462-f001:**
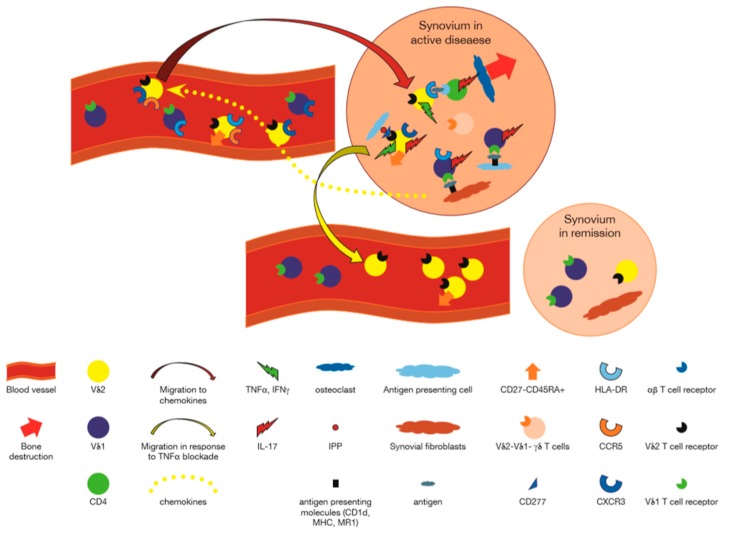
Participation of γδ T cells in Rheumatoid Arthritis. Vδ1^+^ γδ T cells using Vγ3, Vγ8 or other Vγ genes, recognize antigens presented by CD1d or MR1 on synovial fibroblasts, and/or antigens presented by antigen presenting cells, and secrete cytokines such as interleukin (IL)-17, IL-4, and IFNγ [27,37,38,40,41,48]. Chemokines produced in inflamed synovium, attract C-X-C motif chemokine receptor (CXCR)5 and C-C motif chemokine receptor (CCR)3 expressing Vγ9^+^ T cells to the synovium [27]. These cells are activated by phosphoantigens presented by CD277 expressing cells in the synovium to express human leukocyte antigen (HLA)-DR, and in turn, may present antigens to CD4^+^ αβ T cells [24]. Vδ1^−^Vδ2^−^ γδ T cells recognizing unknown antigens also participate in the synovial reaction [35]. IL-17 secreted by CD4^+^ αβ T cells activated by Vγ9^+^ γδ T cells may attract neutrophils and lead to osteoclastogenesis. In the presence of anti tumor necrosis factor (TNF)α antibodies, chemokines retaining the Vγ9^+^ T cells are decreased, and these cells migrate out of the joint to the peripheral blood. Peripheral blood γδ T cells express activation markers acquired in lymph nodes or as a reflection of activation in the synovium [24,25,28].

**Figure 2 cells-09-00462-f002:**
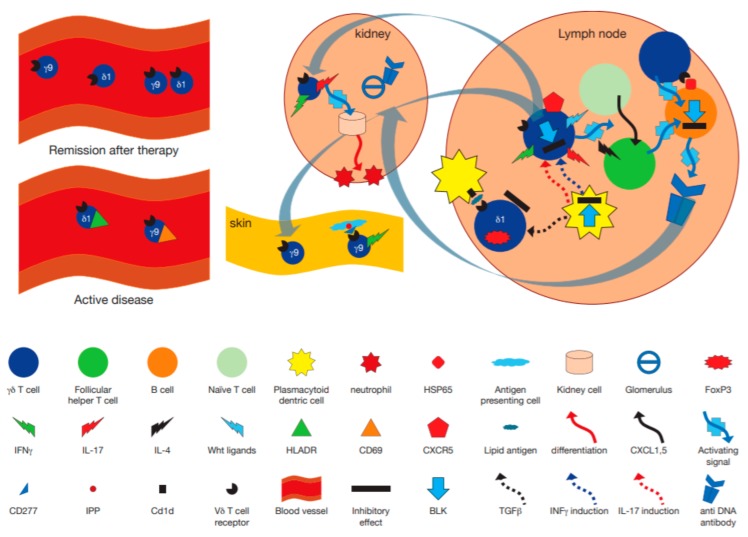
Hypothetical model incorporating the immunopathogenetic role of γδ T cells in human systemic lupus erythematosus (SLE) and murine models. γδ T cells may become activated by plasmacytoid dendritic cells pDC in lymph nodes, leading to their secretion of proinflammatory cytokines such as IL-17 and IFNγ, an activity modulated by BLK [93]. A subset of γδ T cells expressing CXCR5 release Wingless-related integration site (Wnt) proteins, that enhance differentiation of naïve T cells to become follicular helper T cells [100], which in turn, together with IL-4 secretion [97] differentiate B cells to become antigen producing cells making anti DNA antibodies. Other γδ T cells directly interact with heat shock protein (HSP)65 expressing B cells via their T cell receptor (TCR) and help drive anti DNA antibody secretion [92]. At the same time regulatory FoxP3^+^ γδ T cells may become activated by transforming growth factor (TGF)β produced by pDC [18], and by CD1d expressing cells in a TCR dependent manner, to downregulate the immune response [99]. After activation in lymph nodes, γδ T cells could migrate to the kidney where they secrete IL-17, thus enhancing migration of leukocytes [101].

**Figure 3 cells-09-00462-f003:**
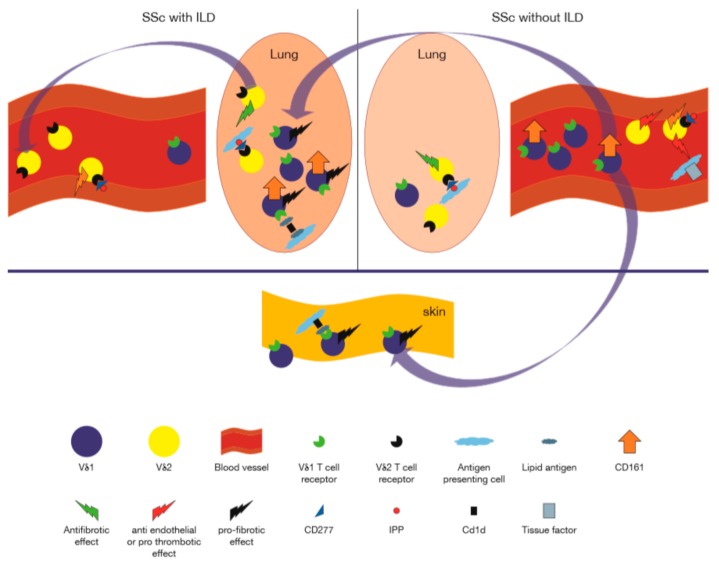
Role of γδ T cells in human systemic sclerosis. On the right, representing patients without interstitial lung disease, are, cytotoxic granzyme expressing Vδ2^+^ γδ T cells in the peripheral blood, which are shown to interact with the endothelium via engagement of the TCR with CD277 endothelial molecules activated by isopentenyl pyrophosphate (IPP), while inducing the procoagulant tissue factor on monocytes, which together could lead to endothelial damage [103,107]. The peripheral blood is enriched in CD161^+^Vδ1^+^ T cells [104]. Profibrotic Vδ1^+^ cells may migrate to the lung, where they encounter cells expressing CD1d in complex with lipids, which trigger Vδ1^+^ T cells to secrete profibrotic factors (e.g., IL-4, CCL3) [104,105,110]. Along with this, exit of Vδ2^+^ T cells (which may potentially confer anti fibrotic functions), from the lung to the peripheral blood takes place [106,111]. These alterations of γδ T cell composition in the lung may contribute to progressive lung disease.

**Table 1 cells-09-00462-t001:** Changes of γδ T cells in autoimmune rheumatic diseases.

Disease (Tissue)	Total γδ T Cells (Relative to Normal)	Vδ1^+^ T Cells (Relative to Normal)	Vγ9Vδ2 T Cells (Relative to Normal)	References
RA (PB)	Equal or decreased	Ratio relative to Vδ2 is increased. Sometimes includes oligoclonal expansions	Equal or decreased (in established long term disease), increased VγδVδ2 TEMRA, decreased naïve Vγ9Vδ2 T cells. Sometimes include oligoclonal expansions. Increases noted after anti TNFα and gold salt therapy. Negative association with disease activity	
RA (synovium)	Polyclonal repertoire sometimes containing oligoclonal expansions common to different joints. HLADR expression increased, CD16 decreased.	Increased relative to Vδ2. Often using Vγ8 or Vγ3 along with Vδ1 in the TCR	Relatively expanded compared to the PB, may use Jδ2.	[32]
JIA (PB)	May be Increased in oligoarticular and quiescent systemic JIA otherwise equal. Increase of IL-17 producers in SJIA	Increase of Vδ1^+^CD69^+^ T cells	Increase of Vδ2^+^CD69^+^ T cells	[71]
JIA (synovium)	Higher than PB in oligoarticular JIA. Otherwise equal to percentage in PB	Higher CD69^+^ than in PB, usually CD45RA^+^, higher in ANA^+^ patients, inversely associated with age at onset, and with recurrence of synovitis	Higher CD69^+^ than PB. Usually CD45RO^+^ inversely associated with age at onset, positively with recovery	[74]
AS (pB)	Total decreased, but enriched for IL23R^+^ γδ T cells secreting IL-17 and in IL-17 and GM-CSF double producing γδ T cells		Elevated in AS patients receiving anti TNFα, secrete IFNγ.	[80,82,83,84]
AS (enthesium/synovium)	RORγt^+^ iNKT and γδ-hi T cells increased, producing IL-17	Enriched for IL-23^+^ RORγt^+^ iNKT and γδ-hi		[80,82,83,84]
SLE (pB)	Decreased, but increase of γδ T cells expressing CD69 and HLADR, and decrease of TNFα and IL-17 secreting cells. Inverse correlation with disease activity. γδ lines help anti DNA production by B cells	decreased, but increased in inactive SLE	decreased	[18,87,88]
SLE (skin)	increased		increased	[90]
SSc [pB)	Decreased especially in early term disease (less than 3 years), diffuse disease and in SCL70^+^ patient. Otherwise equal.	Increased Vδ1^+^ and CD161^+^Vδ1^+^ especially in patients without ILD. Increase of Vδ1^+^CD49d^+^, and HLADR^+^ cells. May be profibrotic in vitro, may respond to cardiolipin via CD1d	Unchanged, decreased, or increased in some patients with ILD, increased granzyme expression, cytotoxic to endothelial cells. Induce fibroblast apoptosis. May be anti fibrotic.	[91]
SSC (skin)		Increased, restricted clonality		[88,90,92,93,94,95]

**Table 2 cells-09-00462-t002:** γδ T cells in animal models of autoimmune rheumatic diseases.

Disease Model	Role of γδ T Cells	References
Rat adjuvant arthritis	No role in disease induction. Possible role in effector phase of disease.	[52,53,54,55]
Murine Collagen induced arthritis	Vγ4/Vδ4^+^ cells producing IL-17 are pathogenic. IL-17 production can be suppressed by inhibitor of RORγt and by IL-28A. ES-62, a phosphorylcholine containing glycoprotein and IL-10 reduce migration of IL-17 producing γδ T cells to the inflamed joint, which are maintained by IL-23, and are not associated with bone destruction.	[58]
Murine BSA induced arthritis	(RORγ)t^+^ IL-17 producing γδ T cells dependent upon IL-23 accumulated in arthritic joints.	[59]
Murine non antigen dependent arthritis	IL-1R and IL-23R expressing Vγ6^+^ γδ IL 17 cells are the main producers of IL-17 in joints of Il1rn ^-^/^-^ mice spontaneously developing arthritis. γδ T cells are responsible for arthritis in B10.RIII mice induced by gene transfer of IL-23. Arthritis induced by intraperitoneal injection of mannan is dependent upon IL-17 secreting γδ T cells.	[67,68,70]
Murine IFNγ^-^knockout (KO)	IL-17 secreting γδ T cells were shown to participate in arthritis and the systemic response to complete Freund adjuvant injection developing in these mice.	[79]
Murine IL-23 gene introduction	increased number of γδ T cells are found in Achilles tendon enthesis, aortic root, and adjacent to the ciliary body and secreted IL-17.	[86]
Murine MRL/lpr model of SLE	γδ T cells are protective from development of glomerulonephritis in the presence of αβ T cells, but mediate a less severe form of disease in their absence, mediated by cytokines and help for B cells. With age, some γδ T cells acquire a CD4^+^B220^+^ phenotype, and produce IL-17. In BLK^+/-^.lpr mice expressing low levels of Bruton lymphocyte kinase gene IL-17 and IFNγ producing γδ T cells are increased enhanced and mediate glomerular damage. γδ T cells induce phosphopeptide P140 mediated apoptosis of lymphocytes, which is associated with amelioration of disease in MRL/lpr mice.	[93,94,97]
lupus-prone NZB/NZW mice	CD1d restricted γδ T cells may be protective in young, and pathogenic in old mice.	[99]
Pristane induced model of SLE	γδ T cells in the kidney expressed IL-17F and A and attracted neutrophils to the kidney. TCRδ^-/-^ mice developed milder glomerulonephritis, due to decreased T follicular helper cell differentiation dependent upon γδ T cell secretion of Wnt ligands.	[101]

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
