# Peer review of "The Role of Gamma Delta T Cells in Autoimmune Rheumatic Diseases"

_cells, 2020, doi:10.3390/cells9020462_

Round 1

Reviewer 1 Report

This rewiev is evidently drawn up with a careful and precise analysis of the literature in which the major ARDs were addressed with reference to γδ T cells. However, the reading appears at times hard to understand, probably due to the large amount of information reported as well as its structuring.

To provide an update of current scientific knowledge, it would be very useful if in the introduction (before examining the role of the γδ T cells in the ARDs) more space is dedicated to the generic description of the discovery, function and mechanism of action of these cells, also increasing the references in this regard (line 34, refs 2 is not exhaustive at the end of that period).

This manuscript should be completely modified for all the figures: they are graphically not easily to read and understand. I strong suggest to make new figures with a dedicate informatic tool.

About all animal models described, it could be a smoother reading to put everything in a summary table

Again, in a comprehensive rewiev, is very usefuful to close the manuscript with a “Concluding Remarks” paragraph.

MINOR REMARKS:

various typo errors should be addressed, as :

line 54: PsA : specify the acronym

the Tab1 Title (Changs= Changes)

Tab 1: pB (capital letters: PB)

Line 123 : typo error

Author Response

I would like to thank the reviewer for his remarks. Here I provide a point by point response. I have addressed all the points raised, I hope successfully. Th e corrected paper has been significantly improved thanks to this process.Reviewer 1. This rewiev is evidently drawn up with a careful and precise analysis of the literature in which the major ARDs were addressed with reference to γδ T cells. However, the reading appears at times hard to understand, probably due to the large amount of information reported as well as its structuring.

To provide an update of current scientific knowledge, it would be very useful if in the introduction (before examining the role of the γδ T cells in the ARDs) more space is dedicated to the generic description of the discovery, function and mechanism of action of these cells, also increasing the references in this regard (line 34, refs 2 is not exhaustive at the end of that period). Response: we have now extended the introduction addressing a generic description of discovery,  function and mechanism of gammadelta T cells and added appropriate references. An additional 12 references or so have been added.

This manuscript should be completely modified for all the figures: they are graphically not easily to read and understand. I strong suggest to make new figures with a dedicate informatic tool.Response: All figures have been modified by a professional scientific graphic artist and are much clearer.

About all animal models described, it could be a smoother reading to put everything in a summary table

Response: A summary table Table 2 has been added.

Again, in a comprehensive rewiev, is very usefuful to close the manuscript with a “Concluding Remarks” paragraph. Response: a concluding remarks section has been added.

MINOR REMARKS:

various typo errors should be addressed, as :

line 54: PsA : specify the acronym corrected

the Tab1 Title (Changs= Changes)  corrected

Tab 1: pB (capital letters: PB) corrected

Line 123 : typo error could not find it

Reviewer 2 Report

Review

The author has addressed the quite hard task of analysing the literature relative to the role of Gamma-Delta T cells in a large spectrum of autoimmune diseases. 

Nevertheless, the review is very exhaustive and well detailed, as well as up to date. On the other end, it is addressed to very experts of the field, and i don’t know if this is the aim of the author and of the journal. In order to make it a little more accessible to a larger public, I would suggest to add a succinct summary at the end of each topic, although the author has done it briefly at some point (also with hard to read sketches). Also, the fluidity of the reading would benefit from a little less details in describing the various studies, and rather stressing the aims of the study and the meaning of the results. 

All together, it is a valuable work and in my modest opinion could only be improved applying the above suggestions. The english is also acceptable, but I think that it might need a review of the punctuation. Moreover, there are  few  points less comprehensible  and are indicated below.

Minor revisions

Pag. 3 lines 131-133  “Thus gamma delta T cells expressing Vgamma 8 …….. synovial tissue of patients with RA”: where is the verb ?

Pag. 5 lines 231-232  “The cells exhibited increased production …… cells correlated with DAS28”: not very clear, please rephrase 

Pag. 10 line 393 JRA or JIA ? Since the authors is reviewing JIA. I know that this nomenclature is often interchangeable, but there is a slight difference.

Pag. 10 lines 396-399 “Similarly, when samples of PB (n = 25) and SF (n = 93) ….percentage of Vδ1+ cells was higher in SF  compared to PB (p< 0.01).”: it is confusing, please rephrase.

Pag. 11 lines 454-457 “Another study confirmed ….that T cells in their PB relative to normal [70].”

Confusing, does it mean that in PB, AS gamma delta were increased respect to the normal controls ?

Author Response

I would like to thank the reviewer for the points raised . I have attempted to address all issues, and responses are indicated point by point. Having done this, I feel the paper is substantially improved, and wish to thank the reviewer for this.

The author has addressed the quite hard task of analysing the literature relative to the role of Gamma-Delta T cells in a large spectrum of autoimmune diseases. 

Nevertheless, the review is very exhaustive and well detailed, as well as up to date. On the other end, it is addressed to very experts of the field, and i don’t know if this is the aim of the author and of the journal. In order to make it a little more accessible to a larger public, I would suggest to add a succinct summary at the end of each topic, although the author has done it briefly at some point (also with hard to read sketches). Also, the fluidity of the reading would benefit from a little less details in describing the various studies, and rather stressing the aims of the study and the meaning of the results. 

Response: we have added summaries at the end of each topic where it was not previously already presented, and provided new professionally prepared figures. We have, throughout the manuscript, attempted to eliminate details and stress aims of studies. All changes are in red.

All together, it is a valuable work and in my modest opinion could only be improved applying the above suggestions. The english is also acceptable, but I think that it might need a review of the punctuation. Response: Punctuation was reviewed and corrected.

Moreover, there are  few  points less comprehensible  and are indicated below.

Minor revisions

Pag. 3 lines 131-133  “Thus gamma delta T cells expressing Vgamma 8 …….. synovial tissue of patients with RA”: where is the verb ? Response:There was a type error. Instead of from it should be form. Then the sentence makes sense. This was corrected.

Pag. 5 lines 231-232  “The cells exhibited increased production …… cells correlated with DAS28”: not very clear, please rephrase.

Response:The sentence was changed: “These cells produced high levels of IL-6, 8 and IFNγ ex-vivo, as well as in vitro, after stimulation with phorbol myristate acetate (PMA) and ionomycin. Furthermore, the number of cytokine producing cells correlated with DAS28” {Guggino, 2018 #125}.

Pag. 10 line 393 JRA or JIA ? Since the authors is reviewing JIA. I know that this nomenclature is often interchangeable, but there is a slight difference.

Response: In order to avoid confusion, all JRA has been reverted to JIA.

Pag. 10 lines 396-399 “Similarly, when samples of PB (n = 25) and SF (n = 93) ….percentage of Vδ1+ cells was higher in SF  compared to PB (p< 0.01).”: it is confusing, please rephrase.

Response: We have clarified by rephrasing : In addition, when samples of PB (n = 25) and SF (n = 93) were analyzed by flow cytometry in 93 JIA patients, Vδ1+ and Vγ9+ γδ T cell subsets were detected in SF of all patients. However, only  the percentage of Vδ1+ cells was higher in SF compared to PB (p< 0.01).  Interestingly, although the distribution of gd T cell subsets was similar in different JIA subgroups, antinuclear antibody (ANA)-positive patients had a higher percentage of SF Vδ1+ T cells than ANA-negative patients (p< 0.01). 

Pag. 11 lines 454-457 “Another study confirmed ….that T cells in their PB relative to normal [70].” 

Confusing, does it mean that in PB, AS gamma delta were increased respect to the normal controls ?

Response: We have clarified this as follows: Another study revealed a decreased percent of gd T cells in the PB of AS patients relative to HC. However, double expressing IL-17+ and  GMCSF+  gd, CD4, and CD8 T cells, and ILC cells are all found in synovium and increased in PB of AS patients .

Round 2

Reviewer 1 Report

The corrected paper has been significantly improved, no other remarks to declare.